X-architecture Steiner minimal tree algorithm based on multi-strategy optimization discrete differential evolution

Liu Genggeng 1
Yang Liliang 1
Xu Saijuan 2
Li Zuoyong 3
Chen Yeh-Cheng 4
http://orcid.org/0000-0001-7668-7425 Chen Chi-Hua 1 chihua0826@gmail.com
1 College of Mathematics and Computer Science, Fuzhou University , Fuzhou , China
2 Department of Information Engineering, Fujian Business University , Fuzhou , China
3 Fujian Provincial Key Laboratory of Information Processing and Intelligent Control, Minjiang University , Fuzhou , China
4 Department of Computer Science, University of California, Davis , Davis, CA , USA
Zhang Qichun
Electronic publication date: 2021 Apr 13
Publication date: 2021
Volume: 7
Electronic Location ID: e473
Received 2021 Feb 11; Accepted 2021 Mar 15
Copyright: © 2021 Liu et al.
Copyright year: 2021
Copyright holder: Liu et al.
License: This is an open access article distributed under the terms of the Creative Commons Attribution License, which permits unrestricted use, distribution, reproduction and adaptation in any medium and for any purpose provided that it is properly attributed. For attribution, the original author(s), title, publication source (PeerJ Computer Science) and either DOI or URL of the article must be cited.
License URL: https://creativecommons.org/licenses/by/4.0/

Keywords: Discrete differential evolution, Steiner minimal tree, Non-Manhattan architecture, Global routing, Multi-strategy optimization

Funding: National Natural Science Foundation of China 61877010 and 11501114 Natural Science Foundation of Fujian Province 2019J01243 Minjiang University MJUKF-IPIC201910 This work was supported by the National Natural Science Foundation of China (Nos. 61877010 and 11501114), the Natural Science Foundation of Fujian Province, China (No. 2019J01243), and the Open Fund Project of Fujian Provincial Key Laboratory of Information Processing and Intelligent Control, Minjiang University (No. MJUKF-IPIC201910). There was no additional external funding received for this study. The funders had no role in study design, data collection and analysis, decision to publish, or preparation of the manuscript.

==============================
Global routing is an important link in very large scale integration (VLSI) design. As the best model of global routing, X-architecture Steiner minimal tree (XSMT) has a good performance in wire length optimization. XSMT belongs to non-Manhattan structural model, and its construction process cannot be completed in polynomial time, so the generation of XSMT is an NP hard problem. In this paper, an X-architecture Steiner minimal tree algorithm based on multi-strategy optimization discrete differential evolution (XSMT-MoDDE) is proposed. Firstly, an effective encoding strategy, a fitness function of XSMT, and an initialization strategy of population are proposed to record the structure of XSMT, evaluate the cost of XSMT and obtain better initial particles, respectively. Secondly, elite selection and cloning strategy, multiple mutation strategies, and adaptive learning factor strategy are presented to improve the search process of discrete differential evolution algorithm. Thirdly, an effective refining strategy is proposed to further improve the quality of the final Steiner tree. Finally, the results of the comparative experiments prove that XSMT-MoDDE can get the shortest wire length so far, and achieve a better optimization degree in the larger-scale problem.

Introduction

At present, very large scale integration (VLSI) technology is developing at a high speed. Initially, the model to solve global routing problem was based on Manhattan structure (Held et al., 2017; Siddiqi & Sait, 2017; Chu & Wong, 2007). There are two ways to connect each pin in this structure, which are horizontal direction and vertical direction. In the development of this structure, limitation of the interconnect wire length optimization appeared, and in the actual situation, there is still a lot of optimization space for wire length of the Steiner minimum tree (SMT). Wire length has a decisive influence on the chip performance. Based on this situation, non-Manhattan structure, which can make full use of the routing resources and shorten the wire length, has become the mainstream model of global routing (Zhu et al., 2020; Zhuang et al., 2020; Zhang et al., 2020b).

X-architecture Steiner minimal tree (XSMT) is a representative model of non-Manhattan structure (Coulston, 2003; Chiang & Chiang, 2002). The SMT problem is to find a minimum connection tree under a given set of pins by introducing additional Steiner points (Liu et al., 2015c). Because SMT cannot be constructed in polynomial time, how to quickly and effectively construct an SMT is a key issue to be solved in VLSI manufacturing process (Liu et al., 2015b, 2019). A heuristic search algorithm has a strong ability to solve NP-hard problem (Liu et al., 2018, 2020a). As a typical heuristic search algorithm, the differential evolution (DE) algorithm has shown good optimization effect in many practical engineering problems. Therefore, based on the DE algorithm, this paper designs relevant strengthening strategies to construct XSMT.

DE is a global optimization algorithm proposed by Storn & Price (1997). Each particle in DE corresponds to a solution vector, and the main process is composed of three steps: mutation, crossover, and selection. DE algorithm has many advantages, such as robustness, reliability, simple algorithm structure and few control parameters, etc., and it has been widely applied in global optimization (Zhao et al., 2020; Ge et al., 2017), artificial intelligence (Zhao et al., 2021; Tang, Zhang & Hu, 2020), bioinformatics (Zhang et al., 2020a), signal processing (Yin et al., 2020; Zhang et al., 2017), machine design (Zhou et al., 2018), and other fields (Ren et al., 2019; Tang et al., 2020b). Generation strategy of trial vector and setting method of control parameters will greatly affect the performance of DE algorithm. Many scholars have improved DE algorithm in these directions, and it has made great progress in recent years. DE was originally proposed for continuous problems and can not be directly used to solve discrete problems such as XSMT; therefore, this paper explores and formulates a discrete differential evolution (DDE) algorithm for solving XSMT problems.

For this reason, this paper proposes a X-architecture Steiner minimal tree algorithm based on multi-strategy optimization discrete differential evolution (XSMT-MoDDE). Firstly, we design an encoding strategy, a fitness function of XSMT, and a population initialization strategy based on Prim algorithm for DDE algorithm to record the structure of XSMT, evaluate XSMT and obtain high quality initial solution, respectively. Secondly, we design an elite selection and cloning strategy, a multiple mutation strategy, and an adaptive learning factor strategy to optimize the search process. At the end of the algorithm, an effective refining strategy is proposed to improve the quality of the final XSMT.

Related work

Research status of RSMT and XSMT

Optimizing the wire length of SMT is a popular research direction, and there are many important research achievements. In Tang et al. (2020a), three kinds of sub problems and three kinds of general routing methods in Steiner tree construction were analyzed, and the research progress in two new technology modes was analyzed (Tang et al., 2020a). Chen et al. (2020a) introduced five commonly used swarm intelligence technologies and related models, as well as three classic routing problems: Steiner tree construction, global routing, and detailed routing. On this basis, the research status of Steiner minimum tree construction, wire length driven routing, obstacle avoidance routing, timing driven routing, and power driven routing were summarized (Chen et al., 2020a). In Liu et al. (2011), Rectilinear Steiner Minimal Tree (RSMT) based on Discrete Particle Swarm Optimization (DPSO) algorithm was proposed to effectively optimize the average wire length (Liu et al., 2011). Liu et al. (2014a) proposed a multi-layer obstacle avoidance RSMT construction method based on geometric reduction method (Liu et al., 2014a). Zhang, Ye & Guo (2016) proposed a heuristic for constructing a RSMT with slew constraints to maximize routing resources over obstacles (Zhang, Ye & Guo, 2016).

Teig (2002) adopted XSMT, which is superior to RSMT in terms of average wire length optimization (Teig, 2002). In Zhu et al. (2005), an XSMT construction method was proposed by side substitution and triangle contraction methods (Zhu et al., 2005). Liu et al. (2020c) constructed a multi-layer global router based on the X-architecture. Compared with other global routers, it had better performance in overflow and wire length (Liu et al., 2020c). Liu et al. (2015c) proposed a PSO-based multi-layer obstacle-avoiding XSMT, which used an effective penalty mechanism to help particles to avoid obstacles (Liu et al., 2015c). In Liu et al. (2020b), a novel DPSO and multi-stage transformation were used to construct XSMT and RSMT. The simulation results on industrial circuits showed that this method could obtain high-quality routing solutions (Liu et al., 2020b). Chen et al. (2020b) proposed an XSMT construction algorithm based on Social Learning Particle Swarm Optimization (SLPSO), which can effectively balance the exploration and exploitation capabilities (Chen et al., 2020b).

The present situation of DE and DDE algorithm

DE algorithm has high efficiency and powerful search ability in solving continuous optimization problems. In the past 20 years after its emergence, many scholars have proposed improved versions of DE algorithm. These improvements better balance the exploitation and exploration ability of DE, and show strong optimization ability on many problems.

An Self-adaptive DE (SaDE) algorithm was proposed in Qin, Huang & Suganthan (2008). In different stages of the evolution process, the value of control parameters is adjusted according to experience, which saves the trial and error cost of developers in the process of adjusting parameters (Qin, Huang & Suganthan, 2008). Rahnamayan, Tizhoosh & Salama (2008) proposed an algorithm for accelerating DE, using opposition-based DE and opposition-based learning methods to initialize population and realize generation jumping to accelerate convergence of DE. Subsequently, Wang, Wu & Rahnamayan (2011) proposed an improved version of accelerated DE, which could be used to solve high-dimensional problems. Wang, Cai & Zhang (2011) proposed Composite DE (CoDE). The algorithm proposed three generation strategies of trial vector and three control parameter settings, and randomly combined the generation strategies and control parameters. The experimental results showed that the algorithm had strong competitiveness (Wang, Cai & Zhang, 2011). Wang, Zeng & Chen (2015) combined adaptive DE algorithm with Back Propagation Neural Network (BPNN) to improve its prediction accuracy.

DDE algorithm is a derivative of DE, which can solve discrete problems. Many existing results have applied DDE algorithm to solve practical problems. In Pan, Tasgetiren & Liang (2008), DDE was used to solve the permutation flow shop scheduling problem with the total flow time criterion. For the total flow time criterion, its performance is better than the PSO algorithm proposed by predecessors (Pan, Tasgetiren & Liang, 2008). In Tasgetiren, Suganthan & Pan (2010), an ensemble of DDE (eDDE) algorithms with parallel populations was presented. eDDE uses different parameter sets and crossover operators for each parallel population, and each parallel parent population has to compete with the offspring populations produced by this population and all other parallel populations (Tasgetiren, Suganthan & Pan, 2010). Deng & Gu (2012) presented a Hybrid DDE (HDDE) algorithm for the no-idle permutation flow shop scheduling problem with makespan criterion. A new acceleration method based on network representation was proposed and applied to HDDE, and the local search of the inserted neighborhood in HDDE was effectively improved to balance global search and local development (Deng & Gu, 2012).

Preliminaries

XSMT problem

Unlike the traditional Manhattan structure, which only has horizontal and vertical connections, two connection methods of 45 and 135 are added to the XSMT problem (Liu, Chen & Guo, 2012; Liu et al., 2015a). This paper introduces the concept of Pseudo-Steiner (PS) point (Definition 1). The PS point exists in two interconnected pins. The fixation of PS point determines the connection method (Definition 2-5) of two pins.

An example of XSMT problem model is as follows. In a given set of pins {p1, p2,…, pn}, pi represents the i − th pin to be connected, and the corresponding coordinate is (xi, yi). Given 5 pins, the corresponding coordinates are shown in Table 1, and the corresponding pin layout is shown in Fig. 1.

Definition 1 Pseudo-Steiner point. Except for pin points, other join points are called Pseudo-Steiner points, denoted as PS points.

Definition 2 Selection 0. As shown in Fig. 2A, draw the vertical edge from A to point PS, and then draw the X-architecture edge from PS to B.

Definition 3 Selection 1. As shown in Fig. 2B, draw the X-architecture edge from A to point PS, and then draw the vertical edge from PS to B.

Definition 4 Selection 2. As shown in Fig. 2C, draw the vertical edge from A to PS, and then draw the horizontal edge from PS to B.

Definition 5 Selection 3. As shown in Fig. 2D, draw the horizontal edge from A to PS, and then draw the vertical edge from PS to B.

Table 1 Coordinate information of pins.

Pin	p1	p2	p3	p4	p5	
Coordinate	(01,22)	(05,05)	(12,10)	(18,03)	(22,16)	

Figure 1 Distribution of pins.

Figure 2 Four selections for connection method.

(A) Selection 0; (B) selection 1; (C) selection 2; (D) selection 3.

Differential evolution algorithm

DE algorithm is a heuristic search algorithm based on modern intelligence theory. The particles of population cooperate and compete with each other to determine the search direction.

The update process of DE

Initialization of the population: N particles are randomly generated, and the dimension of each particle is D. For example, Xi0 represents the particle i, XL is the lower limit of D-dimensional particles, and XH is the upper limit of D-dimensional particles. The corresponding initialization method is as follows:

(1) Xi0=XL+randam(0,1)×(XH−XL)

Mutation operator: In the process of the g-th iteration, mutation operator randomly select three particles Xag, Xbg, and Xcg in the population which are different from each other, and generate particles Vig according to the following mutation formula:

(2) Vig=Xag+F×(Xbg−Xcg)

where F is a learning factor, F ∈ [0,2].

Crossover operator: In the process of crossover, the value of each dimension is selected from Particle Xig or Particle Vig. The probability of selection is cr. The formula of crossover is as follows:

(3) uij={vijrand(0,1)≤crxijelse

where j represents the dimension, cr is the crossover probability, cr ∈ [0,1].

Selection operator: It adopts greedy strategy in the process of selection, that is, selecting the particle with the optimal adaptive value. The formula is as follows:

(4) Xi(g+1)={Vigf(Vig)<f(Xig)Xigelse

where the value of Function f(X) represents the fitness value of Particle X, and the fitness function definitions for each problem are different.

The flow of DE algorithm

Step 1. Initialize the population according to Eq. (1), and initialize the parameters of DE algorithm.

Step 2. Calculate the fitness value of each particle in the population according to fitness function.

Step 3. During each iteration, mutation operation is performed on particles according to Eq. (2) or other mutation operators to produce mutated particles.

Step 4. Check whether the algorithm reaches the termination condition. If so, the algorithm is terminated. Otherwise, return to Step 2 and update the related parameters.

Xsmt-modde algorithm

Encoding strategy

Property 1. The encoding strategy of edge-point pairs is suitable for DDE algorithm, and it can well record the structure of XSMT.

Suppose there are n pin points in the pin graph, and the corresponding Steiner tree has n − 1 edges and n − 1 PS points. Number each pin, determine an edge by recording two endpoints, and add a bit to record selection method of edge. Finally, a bit is added at the end to represent the fitness value of the particle, and the final encoding length is 3×(n−1)+1. The Steiner tree corresponding to pins in Table 1 is shown in Fig. 3, and the corresponding encoding is: 1 3 1 2 3 0 4 5 0 3 4 3 46.284.

Figure 3 Steiner tree.

Fitness function

Property 2. The wire length of XSMT is a key factor that affects global routing results, and the fitness value based on the wire length of XSMT can make the algorithm go in the direction of optimal wire length to the greatest extent.

In an edge set of a XSMT, all edges belong to one of the following four types: horizontal, vertical, 45° diagonal and 135° diagonal. Rotate a 45° diagonal counterclockwise 45° to form a vertical line and a 135° diagonal counterclockwise 45° to form a horizontal line, so that the four types of edges can be replaced by two types. Make the starting point number of all edges smaller than the ending point number, and then sort all edges according to the starting point number, and subtract the overlapping part of the edges. At this time, the total wire length of XSMT can be obtained.

The excellence of XSMT is determined by the total wire length. The smaller the wire length is, the higher the excellence of XSMT will be. Therefore, fitness value measured by XSMT-MoDDE is total wire length of particle. The fitness function of XSMT-MoDDE is shown in Eq. (5).

(5) fitness(Tx)=∑ei∈Txlength(ei)

Initialization

Property 3. Prim algorithm can search an edge subset, which not only includes all the vertices in a connected graph, but also minimizes the sum of the weights of all the edges in subset. Selecting different starting points can get the same weight but different edge subsets. Prim algorithm is used to initialize population, so that particles in population have diversity and the solution space can be reduced at the same time.

Traditional DE algorithm directly uses Eq. (1) to initialize the population. However, for XSMT, if the random strategy is used to initialize each particle (i.e., randomly select a point as root, and use backtracking method to randomly select edges to build a legal tree), will lead to the problem that the solution space is too large to converge well. Therefore, this paper uses Prim algorithm to construct Minimum Spanning Tree (MST) to initialize population. The weight of each edge in MST is determined by Manhattan distance between each two pins. Each particle randomly selects a starting point s to generate a MST and randomly select a connection method for each edge of MST.

The relevant pseudo code is shown in Algorithm 1, where T is edge set of MST, s is starting point, U is point set of MST, V is pin set, P is population, and N is population size. From Lines 1–18 is the function to generate MST. Lines 2–3 randomly select a starting point s and add it to the set U. Line 4 initializes the edge set T. Line 6 selects a visited point i from the set U, and Line 7 sets the minimum cost to infinity. Lines 8–13 select a unvisited point j from the adjacent points of point i, the edge ij with the least cost will be selected and added to set T, and the point j is marked as visited and added to set U. The MST algorithm ends when the set U is the same as the set V, and Line 17 returns a randomly generated MST. Lines 21–24 construct the population, and the initial particle is an MST generated by function PRIMALGORITHM.

Algorithm 1 Initialization strategy based on the Prim algorithm.

Require: V, N	
Ensure: P	
1: function PRIMALGORITHM(V)	
2:   s ← random()/(maxnum+1)×n+1	
3:   U ←{s}	
4:   T ← 0	
5:   while (U! = V) do	
6:     choose point i ∈ U	
7:     mincost ← ∞	
8:     for k ∈ V − U do	
9:       if cost(i, k) < mincost then	
10:         mincost ←cost(i, k)	
11:         j ← k	
12:       end if	
13:     end for	
14:     T ∪ {(i, j)}	
15:     U ∪ {j}	
16:   end while	
17:   return T	
18: end function	
19:	
20: function GENERATEPOPULATION(V, N)	
21:   for i ← 1 to N do	
22:     T ← PRIMALGORITHM(V)	
23:     P ∪ {T}	
24:   end for	
25:   return P	
26: end function	

Elite selection and cloning strategy

Property 4. This strategy proposes two particle mutation strategies based on set, which can mutate elite particles in a very short time. The elite particles are cloned and mutated, and the optimal particle is selected based on greedy strategy to construct a elite buffer with high quality in a short time.

Brief description

The elite selection and cloning strategy consists of four steps: selection, cloning, mutation, and extinction. Part of particles in the population are selected as elite particles, and then the elite particles are cloned to form cloned population. Cloned particles randomly mutate into mutated particles. Mutated particles are selected to enter the elite buffer according to extinction strategy. The elite buffer has the same size as the population and participates in the subsequent process of DE.

The elite selection and cloning strategy can effectively expand the search range of DDE, improve the global search ability of the algorithm, avoid falling into local peaks to a certain extent, and prevent the algorithm from premature convergence.

Algorithm flow

(1) Selection: Sort population according to fitness value, and select the first n particles to form an elite population, n = k × N. k is elite ratio, and the best result can be obtained when k is selected as 0.2 after experimental verification.

(2) Cloning: Clone the particles of the elite population to form a cloned population C. The number of cloned particles is calculated according to Eq. (6).

(6) Ni=round(Ni)

where i is rank of the particle in original population, and round() is rounding down function.

(3) Mutation: The mutation strategy adopts connection method mutation or topology mutation, and two strategies are shown in Fig. 4. Figure 4A shows the mutation process of connection method. The connection method of Line AB is changed from selection 3 to selection 0. Figure 4B shows the mutation process of topology. Line AB is selected to be disconnected and then connected to Line BC. Each cloned particle is assigned to a mutation strategy to form a mutated particle.

Figure 4 Two ways of mutation.

(A) Connection method mutation; (B) topology mutation.

For particles that adopt connection method, randomly select a edges, and the value of a is determined according to the number of edges, as shown in Eq. (7), where n is the number of pins. Then change the connection method of the selected edge.

(7) a=max{1,round(n−110)}

For particles that adopt topology mutation, one edge is randomly disconnected in XSMT to form two sub-XSMTs, and then respectively select a point from the two sub-XSMTs to connect. This process adopts the idea of Disjoint Set Union (DSU) to ensure that a legal tree is obtained after mutation.

(4) Extinction: Select the trial elite particle mbest with the best fitness value in the mutated population. If f(mbest) is better than f(gbest), then mbest will be added to the elite buffer, and all other particles will die, otherwise, all particles in the mutation population will die. If the elite buffer is full, the particle with the worst fitness value will be popped and new particle will be pushed.

The pseudo code of the elite selection and cloning strategy is shown in Algorithm 2, where S represents elite population, M represents mutated population, the inputs are Population P and its size, and the output E represents the elite buffer. Lines 1–9 are selection function, Line 2 calculates the number n of elite particles, Line 3 initializes the Set S, Line 4 establishes a minimum heap according to the fitness value of the population particles, and Lines 5–6 take n elite particles from the top of the minimum heap in turn. Lines 11–28 are the processes of cloning, mutation and extinction. Line 12 initializes Set E, Line 14 initializes Set M, Line 15–20 are cloning and mutation process, Line 15 clones elite particles, Line 16 selects a mutation strategy randomly, and Line 20 adds mutated elite particles to Set M. Lines 22–23 construct two minimum heaps through Set P and Set M. Line 24 compares the tops of the two minimum heaps to determine whether the trial elite particles are saved or died.

Algorithm 2 Elite selection and cloning strategy.

Require: P, N	
Ensure: E	
1: function SELECTION(P)	
2:   n ← k × N	
3:   S ← 0	
4:   H ← heap(P)	
5:   for i ←1 to n do	
6:    S ∪ H.top()	
7:   end for	
8:   return S, n	
9:  end function	
10:	
11: function CLONEMUTATIONANDEXTINCTION(S, n)	
12:   E ← 0	
13:   for i ← 1 to n do	
14:     M ← 0	
15:     for j ← 1 to n/i do	
16:      method ←random(0,1)	
17:      if method == 0 then m ← connection_method_mutation()	
18:      else m ← topology_mutation()	
19:      end if	
20:      M ∪ m	
21:     end for	
22:     H1 ← heap(M)	
23:     H2 ← heap(P)	
24:     if H1.top() < H2.top() then E ∪ H1.top()	
25:     end if	
26:   end for	
27:   return E	
28: end function	

Novel multiple mutation strategy

Property 5. The three novel mutation strategies proposed in this paper introduce the idea of set operations. Under the premise of reasonable computing time, through adjusting edge set of current particle and edge set of other particle, some substructures in XSMT are changed to search for a better combination of substructures.

In DE algorithm, there are six commonly used mutation strategies (Epitropakis et al., 2011), and each strategy uses different basis vectors and differential vectors. The mutation formulas are shown below.

(8) Vig=Xr1g+F(Xr2g−Xr3g)

(9) Vig=Xr1g+F1(Xr2g−Xr3g)+F2(Xr4g−Xr5g)

(10) Vig=Xbestg+F(Xr1g−Xr2g)

(11) Vig=Xbestg+F1(Xr1g−Xr2g)+F2(Xr3g−Xr4g)

(12) Vig=Xig+F(Xbestg−Xig)

(13) Vig=Xr0g+F1(Xbestg−Xr0g)+F2(Xr1g−Xr2g)

where Xrg represents a random particle in population, Xbestg represents the global optimal solution, and F represents learning factor.

Two operating rules

In XSMT-MoDDE algorithm, a particle represents a XSMT. Addition and subtraction operations in the above mutation formulas cannot be directly used in discrete problems. This paper defines two new calculation methods (Definition 6–7).

A is the edge set of particle X1, B is the edge set of particle X2, and the full set is A ∪ B. There are two definitions as follows:

Definition 6 A ⊙ B. ⊙ is expressed as finding the symmetric difference of A and B, which is (A ∪ B) − (A ∩ B), as shown in Fig. 5A.

Figure 5 Operation process of two new operators.

(A) A⊙B; (B) A⊕B.

Definition 7 A ⊕ B. First calculate Set C, C = A − B, and then add the edges of Set B to Set C until Set C can form a legal tree, as shown in Fig. 5B.

Three mutation strategies

In Mutation Strategy 1, basis vector is selected as current particle, and there are two differential vectors. The differential vector of the first stage is generated by the difference between the current particle and the corresponding local historical optimal particle, and Particle T is obtained by Eq. (14). The differential vector in the second stage is generated by the difference between Particle T and the global optimal particle, and target mutated Particle Vig is obtained by Eq. (15).

(14) T=Xig⊕F(Xpbestg⊙Xig)

(15) Vig=T⊕F(Xgbestg⊙T)

In Mutation Strategy 2, basis vector is still current particle, and there are two differential vectors. The differential vector in the first stage is generated by the difference between random particle and the corresponding local historical optimal particle, and Particle T is calculated by Eq. (16). The differential vector in the second stage is generated by the difference between the random particle and global optimal particles, and target mutated Particle Vig is obtained by Eq. (17).

(16) T=Xig⊕F(Xpbestg⊙Xrg)

(17) Vig=T⊕F(Xgbestg⊙Xrg)

In Mutation Strategy 3, basis vector is current particle, and the differential vector is generated by the difference between the current particle and random particle in the population, and the mutated Particle Vig is obtained by Eq. (18).

(18) Vig=Xig⊕F(Xig⊙Xrg)

Mutation Strategy 1 can make particles obtain the partial structure of global optimal particle and the historical local optimal particle, and inherit the characteristics of the two optimal particles, which is a greedy strategy. The implementation of Mutation Strategy 3 can expand the search space and make the mutation direction completely get rid of the structure of the optimal particles, which is suitable for the early stage of iteration and increases the exploration ability of the algorithm. The exploratory ability of Mutation Strategy 2 is between Mutation Strategy 1 and Mutation Strategy 3.

In multiple mutation strategy, the iterative process is divided into two stages by setting a threshold. Three mutation strategies in the early stage are selected with equal probability, and the Mutation Strategy 3 is cancelled in the later stage. The pseudo-code of multiple mutation strategy is shown in Algorithm 3, where P represents population, N represents the size of the population, m represents the number of iterations, t represents threshold, and V represents mutated population. Line 5 judges whether the current iteration is in the early stage of the iteration. If it is in the early stage of the iteration, Mutation Strategy 1, Mutation Strategy 2, and Mutation Strategy 3 are adopted. Line 6 determines whether the current iteration is in the later stage of the iteration. If it is in the latter stage, Mutation Strategy 1 and Mutation Strategy 2 are adopted.

Algorithm 3 Multiple mutation strategy.

Require: P, N, m, e	
Ensure: V	
1: function MUTIMUTATION(P, N, m, t)	
2:   V ← 0	
3:   for i ← 1 to m do	
4:    for j ← 1 to N do	
5:     if i <= t ×N then s ← random(1,2,3)	
6:     else s ← random(1,2)	
7:     end if	
8:     if s == 1 then v ← Mutation1(P[ j])	
9:     else if s == 2 then v ← Mutation2(P[ j])	
10:     else if s == 3 then v ← Mutation3(P[ j])	
11:     end if	
12:     V[ j] ← v	
13:    end for	
14:   end for	
15:   return V	
16: end function	

Adaptive learning factor

Property 6. Learning factor is a key parameter to determine the performance of DDE algorithm, which has a decisive influence on the exploitation and exploration ability of algorithm. This paper proposes an adaptive learning factor based on set operation for the first time to effectively balance the search ability of XSMT-MoDDE algorithm.

Operating rule for learning factors

As shown in Eq. (2), the learning factor F acts on the difference vector and controls the global search capability of DDE algorithm (Wang et al., 2014; Gong et al., 2010; Brest et al., 2006). In discrete problems, simple multiplication operation cannot be used. This paper redefines the * operation in Eq. (2).

Definition 8 F∗(Xbestg⊙Xrg) F < 1. Randomly eliminate n edges {e1, e2,…, en} from the edge set of difference particles, where ei∈Xbestg and ei∉Xig, and the value of n is calculated by Eq. (19).

Definition 9 F∗(Xbestg⊙Xrg) F > 1. Randomly eliminate n edges {e1, e2,…, en} from the edge set of difference particles, where ei∈Xig and ei∉Xbestg, and the value of n is calculated by Eq. (20).

Definition 10 F∗(Xbestg⊙Xrg) F = 1. No changes are made to the edge set.

(19) n=round((1−F)×|Xbestg|)

(20) n=round((F−1)×|Xig|)

where |X| represents the number of edge of Particle X.

Adaptive update process

Each Particle Xi corresponds to the adaptive learning factor Fi, which is initialized to 1. After each selection operation, the Parameter Fi is updated.

(1) Calculate reference Parameter r, r k× fbest + 1, where k is 0.001 and fbest is the fitness value of the global optimal particle;

(2) Calculate difference value δ between fitness value fi of Xig and fitness value fbest of Xbestg;

(3) Update Fi, the update formula is as follows:

(21) Fi={Fi+0.05Δ>rFi−0.05Δ≤r

When the fitness value fi is close enough to fbest, reduce Fi to preserve its structure to a greater extent, otherwise, increase Fi to expand the global search capability.

Refining strategy

Property 7. Refining strategy minimizes wire length of XSMT under the determined topology within a reasonable time.

There may still be space for optimization for the optimal particles at the end of iteration. In order to search for a better result, a refining strategy is proposed. The steps of algorithm are as follows:

(1) Calculate degree of each Point pi in the optimal particle. The degree is defined as the number of edges connected to point, denoted as di;

(2) There are 4 kinds of edges in X-architecture. If the degree of Point pi is di, there are 4di types of substructures corresponding to the point. The set of all substructures corresponding to Point pi is S, and edge Set E is obtained when the substructures corresponding to Points p1 − pi − 1 have been determined. Calculate common wire length l between Substructure si in Set S and Set E, select Substructure si corresponding to the largest l, and add the edges of si to the Set E. The algorithm ends until all points have been visited.

The pseudo code of the refining strategy algorithm is shown in Algorithm 4, where X represents the target particle obtained by the XSMT-MoDDE algorithm, n represents the point number of XSMT, and R represents the refined particle. Line 2 initializes Set R. Lines 3-20 search for the optimal substructure corresponding to each point. Line 4 calculates the degree of Point pi, Line 5 initializes maximum common wire length, and Line 6 initializes the optimal substructure set. Lines 7–14 calculate common wire length and update the largest common wire length. Lines 15–19 store the edges in the optimal substructure into Set R.

Algorithm 4 Refining strategy.

Require: X, n	
Ensure: R	
1: function REFINING(X, n)	
2:  R ← 0	
3:  for i ← 1 to n do	
4:   d ← CalculateDegree(Xi)	
5:   Length ← 0	
6:   Substructure ← 0	
7:   for j←1 to 4d do	
8:    s ← GetSubstructure()	
9:    l ← GetCommonWireLength()	
10:    if l > Length then	
11:      Substructure ← s	
12:      Length ← l	
13:    end if	
14:   end for	
15:   for edge in Substructure do	
16:   if edge not in R then	
17:    R ∪ edge	
18:   end if	
19:   end for	
20:  end for	
21:  return R	
22: end function	

Related parameters

The main parameters of the algorithm in this paper include population size n, iteration times m, threshold t, learning factor F, and crossover probability cr.

In the proposed algorithm, n is 50, m is 500, and t is 0.4. The adaptive strategy of learning factor F has been described in detail in Section 3.6. The crossover probability cr also adopts the adaptive strategy, which is as follows:

(22) cri={crl+(cru−crl)fi−fminfmax−fminfi>f¯crlelse

where cri = 0.1, cru = 0.6, fi represents the fitness value of the current particle, fmin represents the minimum historical fitness value, fmax represents the maximum historical fitness value, and f¯ represents the average historical fitness value.

The algorithm flow of XSMT-MoDDE

The algorithm flow chart of XSMT-MoDDE is shown in Fig. 6, and the detailed flow is as follows:

Initialize threshold, population size, adaptive learning factor F, and adaptive crossover probability cr.

Use Prim algorithm to construct initial particles and generate initial population.

Check the current stage: early stage or late stage of iteration.

Select a mutation strategy from the corresponding mutation strategy pool according to the current stage. Obtain the mutated particles according to the mutation strategy.

Obtain the trial particles according to the crossover operator.

Obtain the next generation of particles according to the selection operator.

Adopt elite selection and cloning strategy, and update the elite buffer after four steps of selection, clone, mutation, and extinction.

Update adaptive learning factor and adaptive crossover probability by Eqs. (21) and (22).

Check the number of iterations, and end the iteration if the termination condition is met, otherwise, return to Step (3).

At the end of XSMT-MoDDE algorithm, refining strategy is adopted to obtain the target solution.

Figure 6 Algorithm flowchart.

Complexity analysis of XSMT-MoDDE algorithm

Property 8. When the population size is m and the number of pins is n, the time complexity of one iteration is O(mnlogn).

Complexity analysis of multiple mutation operator

The mutation process is divided into two stages. First, difference vector is constructed, and then difference vector and the basis vector are used to construct the trial particles.

Construction of difference vector: Sort the edges of two edge sets according to the number of edge start point, and use binary search to construct the non-common edges. The complexity of this process is O(nlog(n)), and the non-common edge set is the difference vector.

Construction of mutation particle: Construct the difference set of basis vector and difference vector according to the above-mentioned similar idea. Then the edges in the difference set are stored in DSU, and edges are randomly selected from difference vector to be added to DSU until a complete tree is constructed. The time complexity of this process is O(nlog(n)).

Complexity analysis of elite selection and cloning strategy

A minimum heap is established according to the fitness value of particles, and the heap top is selected for cloning each time. The time complexity required for this process is O(n).

The mutation process adopts connection method mutation and topology mutation. The connection method mutation selects two different edges randomly from the edge set to modify the connection method of the edges. The time complexity required is O(1). In topology mutation, one edge is randomly disconnected to form two sub-XSMTs, which are recorded using the DSU. It takes O(nlog(n)) time to construct two sub-XSMTs with DSU, and randomly select one point from each of two sub-XSMTs to establish connection, this process takes O(1) time.

The particles obtained by the elite selection and cloning strategy need to be stored in an elite buffer with a size of m. The population particles and the particles of elite buffer participate in mutation, crossover, and selection operations together.

Complexity analysis of refining strategy

The degree of Point i is recorded as di. We always keep di within 4, even if there is a minimum probability greater than 4, only four connected edges will be considered in refining strategy. The adjacent edges of a point select a connection method respectively to form a substructure. An X-architecture edge has four selection methods, so one point corresponds to 4di substructures, where 4di ≤ 256.

Refining strategy takes out the optimal particle constructed by XSMT-MoDDE algorithm, enumerates substructures for each point of the particle, and obtain the substructure with the largest common wire length. So for the case of n points, the required time is ∑i=1n(di×4di).

Experimental results

The proposed XSMT-MoDDE has been implemented in C++ language on a windows computer with 3.5 GHz Intel CPU. To compare the experimental results fairly, we run all programs in the same experimental environment and use the same benchmarks from GEO and IBM. The population size and iteration size of all heuristic algorithms are set to 50 and 500 respectively. Calculation formula of optimization rate is shown in Eq. (23).

(23) rate=b−ab×100%

where a is the experimental result of the XSMT-MoDDE algorithm, and b is the experimental result of other algorithms.

Verify the effectiveness of multi-strategy optimization

Experiment 1: In order to verify the effectiveness of the multi-strategy optimization DDE algorithm in constructing XSMT, this experiment will compare the results of XSMT-MoDDE algorithm and XSMT-DDE algorithm. Experimental results are shown in Tables 2 and 3. Table 2 is the optimization results of wire length, and Table 3 is the optimization results of standard deviation. The results show that multi-strategy optimization can achieve an average wire length optimization rate of 2.35% and a standard deviation optimization rate of 95.69%. This experiment proves that multi-strategy optimization has a powerful effect on wire length reduction, and at the same time greatly increases the stability of DDE.

Table 2 Average wire length optimization results of multi-strategy optimization.

Circuit	Pins	XSMT-DDE	XSMT-MoDDE	Reduction (%)	
1	8	16,956	16,900	0.33	
2	9	18,083	18,023	0.33	
3	10	19,430	19,397	0.17	
4	15	25,728	25,614	0.44	
5	20	32,434	32,171	0.81	
6	50	49,103	48,090	2.06	
7	70	57,386	56,397	1.72	
8	100	70,407	68,917	2.12	
9	400	145,183	139,871	3.66	
10	410	146,680	141,571	3.48	
11	500	160,031	154,406	3.51	
12	1,000	232,057	220,577	4.95	
Average				1.97	

Table 3 Standard deviation optimization results of multi-strategy optimization.

Circuit	Pins	XSMT-DDE	XSMT-MoDDE	Reduction (%)	
1	8	56	0	100.00	
2	9	58	0	100.00	
3	10	42	0	100.00	
4	15	198	10	94.95	
5	20	343	51	85.13	
6	50	1,036	147	85.81	
7	70	1,082	102	90.57	
8	100	1,905	279	85.35	
9	400	3,221	120	96.27	
10	410	3,222	178	94.48	
11	500	3,193	139	95.65	
12	1,000	3,977	106	97.33	
Average				93.80	

Verify the effectiveness of refining strategy

Experiment 2: In order to verify the effectiveness of the refining strategy, this experiment will compare the results of refined XSMT-MoDDE algorithm and XSMT-MoDDE algorithm. The experiment result is shown in Tables 4 and 5. Table 4 is the optimization results of wire length, and Table 5 is the optimization results of standard deviation. The results show that refining strategy can achieve an average wire length optimization rate of 0.50% and a standard deviation optimization rate of 37.30%. From the experimental results and the above complexity analysis, it can be seen that after XSMT-MoDDE algorithm is over, refining strategy only takes a short time to obtain a lot of optimization of wire length and standard deviation. Regardless of whether refining strategy is added or not, both can always obtain accurate solutions in circuits with less than 10 pins. Refining strategy has more significant optimization effects in larger circuits.

Table 4 Average wire length optimization results of refining strategy.

Circuit	Pins	XSMT-DDE	Refining	Reduction (%)	
1	8	16,900	16,900	0.00	
2	9	18,023	18,023	0.00	
3	10	19,397	19,397	0.00	
4	15	25,614	25,624	−0.04	
5	20	32,171	32,091	0.25	
6	50	48,090	48,090	0.00	
7	70	56,397	56,105	0.52	
8	100	68,917	68,457	0.67	
9	400	139,871	138,512	0.97	
10	410	141,571	140,359	0.86	
11	500	154,406	152,649	1.14	
12	1,000	220,577	217,060	1.59	
Average				0.50	

Table 5 Standard deviation optimization results of refining strategy.

Circuit	Pins	XSMT-DDE	Refining	Reduction (%)	
1	8	0	0	–	
2	9	0	0	–	
3	10	0	0	–	
4	15	10	8	20.00	
5	20	51	22	56.86	
6	50	147	119	19.05	
7	70	170	136	20.00	
8	100	279	187	32.97	
9	400	120	57	52.50	
10	410	178	56	68.54	
11	500	139	50	64.03	
12	1,000	115	113	1.74	
Average				37.30	

Algorithm comparison experiment

Experiment 3: To compare the performance of XSMT-MoDDE algorithm with other heuristic algorithms, we compare the results of XSMT constructed by MoDDE algorithm, DDE algorithm, Artificial Bee Colony (ABC) algorithm, and Genetic Algorithm (GA). The experimental results are shown in Tables 6, 7, and 8. XSMT-MoDDE compares with XSMT-DDE, XSMT-ABC, and XSMT-GA, the average wire length is reduced by 2.40%, 1.74%, and 1.77%, the optimal wire length is reduced by 1.26%, 1.55%, and 1.77%, and the standard deviation is reduced by 95.65%, 33.52%, and 28.61%. Experimental results show that XSMT-MoDDE is better than XSMT-DE, XSMT-ABC, and XSMT-GA in both the wire length and standard deviation indicators. Compared with other algorithms, this algorithm still has excellent stability on the basis of having better wire length results.

Table 6 Comparison results of average wire length in the GEO dataset.

Circuit	Pins	Mean value	Reduction (%)	
DDE	ABC	GA	MoDDE	DDE	ABC	GA	
1	8	16,956	16,918	16,918	16,900	0.33	0.00	0.00	
2	9	18,083	18,041	18,041	18,023	0.33	0.10	0.10	
3	10	19,430	19,696	19,696	19,397	0.17	1.52	1.52	
4	15	25,728	25,919	25,989	25,624	0.40	1.14	1.40	
5	20	32,434	32,488	32,767	32,091	1.06	1.22	2.06	
6	50	49,103	48,940	48,997	48,090	2.06	1.74	1.85	
7	70	57,386	57,620	57,476	56,105	2.23	2.63	2.39	
8	100	70,407	70,532	70,277	68,457	2.77	2.94	2.59	
9	400	145,183	141,835	141,823	138,512	4.59	2.40	2.40	
10	410	146,680	143,642	143,445	140,359	4.31	2.29	2.15	
11	500	160,031	156,457	156,394	152,649	4.61	2.43	2.39	
12	1,000	232,057	222,547	222,487	217,060	5.90	2.47	2.44	
Average						2.40	1.74	1.77	

Table 7 Comparison results of best wire length in the GEO dataset.

Circuit	Pins	Best value	Reduction (%)	
DDE	ABC	GA	MoDDE	DDE	ABC	GA	
1	8	16,918	16,918	16,918	16,900	0.11	0.11	0.11	
2	9	18,041	18,041	18,041	18,023	0.10	0.10	0.10	
3	10	19,415	19,696	19,696	19,397	0.09	1.52	1.52	
4	15	25,627	25,627	25,897	25,605	0.09	0.09	1.13	
5	20	32,209	32,344	32,767	32,091	0.37	0.78	2.06	
6	50	47,987	48,637	48,783	47,975	0.03	1.36	1.66	
7	70	56,408	57,227	57,445	55,919	0.87	2.29	2.66	
8	100	68,829	70,382	70,092	68,039	1.15	3.33	2.93	
9	400	141,967	141,490	141,467	138,382	2.53	2.20	2.18	
10	410	144,033	143,310	143,282	140,179	2.68	2.18	2.17	
11	500	156,950	156,034	156,110	152,591	2.78	2.21	2.25	
12	1000	226,654	222,262	222,285	216,824	4.34	2.45	2.46	
Average						1.26	1.55	1.77	

Table 8 Comparison results of standard deviation in the GEO dataset.

Circuit	Pins	Standard deviation	Reduction (%)	
DDE	ABC	GA	MoDDE	DDE	ABC	GA	
1	8	56	0	0	0	100.00	–	–	
2	9	58	0	0	0	100.00	–	–	
3	10	42	0	0	0	100.00	–	–	
4	15	198	148	46	8	95.96	94.59	82.61	
5	20	343	118	45	22	93.59	81.36	51.11	
6	50	1,036	242	133	119	88.51	50.83	10.53	
7	70	1,082	195	140	136	87.43	30.26	2.86	
8	100	1,905	69	112	187	90.18	−171.01	−66.96	
9	400	3,221	200	170	57	98.23	71.50	66.47	
10	410	3,222	146	122	56	98.26	61.64	54.10	
11	500	3,193	160	133	50	98.43	68.75	62.41	
12	1,000	3,977	131	107	113	97.16	13.74	−5.61	
Mean						95.65	33.52	28.61	

Experiment 4: In the stage of global routing, there are tens of thousands of nets on the circuit board, and pins inside net need to be interconnected. This paper uses XSMT-MoDDE algorithm to optimize wire length of global routing. This experiment adopts the benchmark provided by IBM, and XSMT-MoDDE algorithm, SAT algorithm, and KNN algorithm are used to construct XSMT. The experimental results are shown in Table 9. Compared with SAT and KNN, XSMT-MoDDE optimizes wire length by 10.05% and 8.86% respectively. Experimental results show that XSMT-MoDDE can greatly shorten the wire length in the construction of multi-nets XSMT problem, and provide effective guidance for global routing.

Table 9 Comparison results of wire length in the IBM dataset.

Circuit	Nets	Pins	Value	Reduction (%)	
SAT	KNN	MoDDE	SAT	KNN	
ibm01	11,507	44,266	61,005	61,071	56,080	8.07	8.17	
ibm02	18,429	78,171	172,518	167,359	154,868	10.23	7.46	
ibm03	21,621	75,710	150,138	147,982	133,999	10.75	9.45	
ibm04	26,263	89,591	164,998	164,828	149,727	9.26	9.16	
ibm06	33,354	124,299	289,705	280,998	256,674	11.40	8.66	
ibm07	44,394	164,369	368,015	368,015	335,556	8.82	8.82	
ibm08	47,944	198,180	431,879	413,201	371,948	13.88	9.98	
ibm09	53,039	187,872	418,382	417,543	382,282	8.63	8.44	
ibm10	64,227	269,000	588,079	589,102	532,644	9.43	9.58	
Mean						10.05	8.86	

Finally, for a better understanding the results of XSMT-MoDDE algorithm, we use Matlab to simulate the final XSMT diagrams. We choose Circuit 11 and Circuit 12 in Table 7 as representatives, as shown in Figs. 7A and 7B.

Figure 7 Steiner tree generated by XSMT-MoDDE.

(A) Steiner tree with 500 pins; (B) Steiner tree with 1,000 pins.

Conclusions

This paper designs four optimization strategies. The first three optimization strategies are used to strengthen DDE algorithm, and the fourth optimization strategy is used to reduce the wire length of final particle to the greatest extent.

Elite selection and cloning strategy expands the search range and enhances the diversity of the population particles. The elite particles are cloned and mutated, and the most excellent particle is selected greedily. This strategy enables the algorithm to quickly converge to a better state. Novel multi-mutation strategy introduces the idea of set operation. Through the interaction between edge sets, the corresponding shape of XSMT is changed. Three mutation strategies have different exploitation and exploration capabilities, and the three strategies are used alternately to avoid the algorithm from converging to the local peak prematurely. Adaptive learning factor dynamically adjusts and retains the ratio between the current particle edge set and the optimal particle edge set. Effectively improve global exploitation and local exploitation capabilities, and seek a balance between random strategy and greedy strategy.

The XSMT-MoDDE algorithm proposed in this paper uses three indicators to measure algorithm results which are average wire length, optimal wire length, and standard deviation as evaluation. The proposed algorithm has achieved better optimization results compared with other algorithms. Moreover, XSMT-MoDDE has a stronger optimization ability in circuits with large-scale circuits. It is better than the results of the SAT and KNN algorithms in the case of multi-nets. Therefore, the XSMT-MoDDE algorithm has good application prospect in the stage of global routing. In the future, we will study the construction of obstacle avoidance XSMT by multi-strategy optimization DDE.

Additional Information and Declarations

Competing Interests

Author Contributions

Data Availability

Chi-Hua Chen is an Academic Editor for PeerJ Computer Science.

Genggeng Liu conceived and designed the experiments, performed the experiments, analyzed the data, performed the computation work, prepared figures and/or tables, and approved the final draft.

Liliang Yang conceived and designed the experiments, performed the experiments, analyzed the data, performed the computation work, prepared figures and/or tables, and approved the final draft.

Saijuan Xu conceived and designed the experiments, performed the experiments, analyzed the data, performed the computation work, prepared figures and/or tables, and approved the final draft.

Zuoyong Li conceived and designed the experiments, performed the experiments, authored or reviewed drafts of the paper, and approved the final draft.

Yeh-Cheng Chen analyzed the data, authored or reviewed drafts of the paper, and approved the final draft.

Chi-Hua Chen performed the computation work, authored or reviewed drafts of the paper, and approved the final draft.

The following information was supplied regarding data availability:

The source codes and data files are available at GitHub:

https://github.com/yll7960/XSMT_MoDDE.

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
