# Peer review of "X-architecture Steiner minimal tree algorithm based on multi-strategy optimization discrete differential evolution"

_PeerJ Computer Science, doi:10.7717/peerj-cs.473_

## Round 0.1 · original submission · Minor Revisions

Based on the reviewers' comments, a minor revision is needed to improve the quality of the paper. In particular, the presented algorithm should be further explained in terms of implementation. The flow of the algorithm should be given in detail. In addition, the results should be compared with existing algorithms and some figures in the paper should be further explained as well to improve the readability of the manuscript.

Reviewer 1 ·

Basic reporting

This paper proposes an X-architecture Steiner minimal tree algorithm based on multi-strategy optimization discrete differential evolution. The layout of the article conforms to the norms, the language is fluent, and no grammatical errors have been found.

Experimental design

The experimental design of this paper is complete and the experimental comparison is rich, and the experimental results well reflect the optimization degree of average wire length, optimal wire length, and standard deviation. Finally, the experimental part well reflects the effectiveness of the proposed algorithm.

Validity of the findings

The proposed algorithm can effectively reduce the wire length of X-architecture Steiner minimal tree and can obtain the best results currently.

Additional comments

The construction of Steiner tree is a hot topic in related fields, and I give relevant revision opinions after reading this paper.
(1) The flow of XSMT-MoDDE algorithm can be more detailed.
(2) What is the future research work on this work? It is suggested to give future research work in Conclusion Section of the paper.

Reviewer 2 ·

Basic reporting

Global routing is an important link in Very Large Scale Integration (VLSI) design. As the best model of global routing, X-architecture Steiner Minimal Tree (XSMT) has a good performance in wire length optimization. For this reason, an X-architecture Steiner Minimal Tree algorithm based on Multi-strategy optimization Discrete Differential Evolution (XSMT-MoDDE) is proposed. This work proposes four strategies to optimize the DDE algorithm. The four strategies are somewhat innovative. As a whole, this paper has fluent grammar and rigorous logic, and has designed a wealth of experiments to verify the results.
After reading the paper, I give some additional revised opinions for reference: Firstly, are these optimization strategies only used for DDE algorithms, and can they be used by other evolutionary algorithms? Secondly, the mutation process in Figure 4 can be further explained. Thirdly, further investigation and citation for this paper are recommended for relevant references including VLSI routing.

Experimental design

This paper has verification experiments and comparative experiments, and the experimental design is more reasonable. Results of the comparative experiments prove that XSMT-MoDDE can get the shortest wire length so far, and achieve the better optimization degree in the larger-scale problem.

Validity of the findings

Under the premise of reasonable time complexity, the constructed XSMT obtains the current optimal wire length which is the most important objective of Steiner Minimal Tree.

Additional comments

none

---

## Round 0.2 · accepted · Accept

Both reviewers are satisfied with the revised version where all the comments have been addressed well. Both reviewers recommend accepting the current version as it is, which leads to the final decision.

Reviewer 1 ·

Basic reporting

no comment

Experimental design

no comment

Validity of the findings

no comment

Additional comments

This work proposes four strategies to optimize the DDE algorithm. Results of the comparative experiments prove that XSMT-MoDDE can get the shortest wire length. The paper writing is fluent.

Reviewer 2 ·

Basic reporting

The authors explained all the problems I have mentioned and revised the paper carefully, so I do not have any more question.

Experimental design

I think it is sufficient and this work is well written.

Validity of the findings

The problem of this work is well presented and make sense.

Additional comments

The authors explained all the problems I have mentioned and revised the paper carefully, so I do not have any more question.